# Removing the Scaling Error Caused by Allometric Modelling in Forest Biomass Estimation at Large Scales

**Carl Zhou [1] and Xiaolu Zhou [2,3,*]**

[1] Faculty of Health Sciences, University of Ottawa, Ottawa, ON K1N 6N5, Canada
[2] Research Center for Ecological Forecasting and Global Change, Northwest A&F University, Yangling 712100, China
[3] Ecological Modeling and Carbon Science, Department of Biology Science, University of Quebec at Montreal, Montreal, QC H3C 3P8, Canada
[*] Correspondence: xiaolu.zhou@nwafu.edu.cn or zhoux1977@163.com

**Abstract:** To estimate the responses of forest ecosystems, most relationships in biological systems are described by allometric relationships, the parameters of which are determined based on field measurements. The use of existing observed data errors may occur during the scaling of fine-scale relationships to describe ecosystem properties at a larger ecosystem scale. Here, we analyzed the scaling error in the estimation of forest ecosystem biomass based on the measurement of plots (biomass or volume per hectare) using an improved allometric equation with a scaling error compensator. The efficiency of the compensator on reducing the scaling error was tested by simulating the forest stand populations using pseudo-observation. Our experiments indicate that, on average, approximately 94.8% of the scaling error can be reduced, and for a case study, an overestimation of 3.6% can be removed in practice from a large-scale estimation for the biomass of *Pinus yunnanensis* Franch.

**Keywords:** aggregation error; allometric equation; error compensation; scaling error; variable allometric ratio

## 1. Introduction

Most relationships in biological systems are nonlinear between two organs, subsystems, and groups [1]. These relationships are usually described as allometric [2–4] and their parameters can be determined by regression analysis based on observations [5,6]. That is, our ecological knowledge mostly comes from small, easily measured sample plots [7,8]. These plot observations and estimates can be extrapolated to large-scale estimates [9] and global changes [10,11]. The technology of upscaling using allometric relationships was developed gradually. Originally, the allometry theory was used to study individual living bodies, e.g., single tree, and the relative growth of their different organs [2]. This theory has been widely applied in various fields, from biology to sociology [12,13]. In the research field of forest ecosystems, allometric equations were further extended for depicting the relationship between different tree parts in order to address various issues [5,14,15]. Since the 1980s researchers have compiled several large databases across the world [16–18] according to field measurements of forest biomass which had been converted from tree level to plot level by measurers. These hectare-based data provide the opportunity to test allometric relationships at both the plot-scale and larger scales. Using this methodology of model testing, the analysis in our study was carried out based on plot measurements (biomass or volume per hectare) rather than individual trees—that is, to extrapolate biomass from plot-scale to large-scale.

When describing the properties of a large-scale ecosystem using existing fine-scale knowledge, the scaling error occurs during the transforming processing [19–21] (also see Appendix A). This causes prediction bias on large-scales [20] because of the method of weighted mean [19,22]. Specifically, during the scaling of ecological properties from field plots up to a region, the variability among the plots can be aggregated [23] by using a nonlinear equation that only reflects the behavior of the plots [23–25]. The scaling errors usually include two types that respectively correspond to two stages in forest biomass estimation. The first stage is volume prediction, which is carried out based on large sampling and hierarchical data structure in national forest inventories (NFI) [26], in which the sampling error would exist [27]. Following it, the second stage converts forest biomass from volume information (m$^3$ ha$^{-1}$) released in NFI reports. As a simple conversion, the biomass expansion factor (BEF) and biomass conversion expansion factor (BCEF) are generally used to estimate forest biomass [28]. For more accurate estimations, the volume-based biomass equations were also frequently employed in past decades [29–34]. This requires removing the scaling error at the regional level. Our study focuses on the second stage and second scaling error. This stage is implemented especially for those regions or countries that have NFI without biomass data. In this stage, the scaling error is not involved with trees because upscaling is only based on the means per unit area.

According to the computational principle, the form of the allometric equation affects the magnitude of scaling error [23]. Zhou et al. [25] have reported a quantitative analysis to reduce scaling error caused by power-law equations. However, besides the general power-law equation, an improved allometric equation [35] has not been addressed. The improved equation incorporates a variable allometric ratio (VAR) instead of a constant ratio. It is more complex in form than conventional power-law equations. Because it can explain statistically significant variation and give well-fitting curves according to field measurements [35,36] for some special species [37], this equation increases the accuracy of biomass estimates, particularly for the species (*Pinus yunnanensis* Franch.) addressed in our study. Our analysis adopted the improved allometric equation and presented a quantitative approach on the issues of scaling error and error compensation. The presented method provides guidance for statistical and mechanistic analyses of errors for scaling forest biomass from fine-scale to large-scales.

Our objectives are to test the margin of scaling error, to derive a user-friendly compensator, and to examine its ability in correcting scaling error in the application of the improved allometric equation for large-scale estimation of forest ecosystem biomass. We only focus on correcting the scaling error itself and leave out other types of errors in the biomass estimation.

## 2. Methods

### 2.1. Derivation

Ruark et al. [35] presented an improved allometric VAR equation $y = rx^k e^{-ux}$ where y denotes total biomass density (t ha$^{-1}$), x is stand stocking volume (m$^3$ ha$^{-1}$), and *r*, *k*, and *u* are parameters greater than zero. The error compensator for this equation is derived by expanding the scaling error expression (*error* = $Y_1 - Y_2$, see Appendix A) as Taylor series, and utilizing the central moment method to solve the items in Taylor series. Eventually the difference of scaling error before and after compensation is proved by an analytical expression. This expression includes two parameters that are used in the allometric equation. The variables in the expression reflect the feature of the estimated forest ecosystem, i.e., nonlinearity of volume-to-biomass relationship.

### 2.2. Simulation

To test the algorithm and efficiency of the error compensator, we construct a pseudo population, and compare our biomass estimate with the true value of the population. The error compensation is simulated by scaling tree biomass from stand level up to a large region consisting of 100 stands with different stages of development (see Supporting Information). The area and volume of the 100 stands are produced by pseudo-random number generator under uniform probability distribution. The area

ranges from 1.0 to 10.0 ha and stand stocking volume ranges from 15 to 300 (m$^3$ ha$^{-1}$). Thus, the true value of the total amount of regional forest biomass is recognized by accumulating all stand biomasses. Based on the pseudo population, numerous trials provide a uniform distribution that gives how much error can be reduced and information on how many times the convergence of compensated percentage will be reached.

*2.3. Data for Case Study*

A case study of forest biomass estimation is carried out to test the difference that could come from scaling error at the provincial scale. We estimated the biomass of *Pinus yunnanensis* in Yunnan province of China. This province is located in the southwest border of China, between latitude 21°8′32″–29°15′8″ N and longitude 97°31′39″–106°11′47″ E. The total forested area and stem volume for the species is 2.92 million ha and 222.43 million m$^3$ in the province [38]. The data of area and volume were classified as different age groups: young, middle, near-mature, mature, and over-mature. The range of mean stocking volume was estimated according to a literature review [38–40]. The variance of stocking volume is computed as the between-group variance (refer to 3.3. Comparison), yet the within-group variance could not be computed because the detailed data are not released to public in China. The field measurements used in the case study were collected from a published database [18]. This database compiled 34 plot measurements reported by 24 studies that carried out field studies in different years. Due to the costly destructive measurement of forest biomass, each of the studies only addressed one or a few plot works. All measurements have been scaled from tree-level to plot-level and hectare-based values by data measurers. All biomass values are measured by oven-drying different parts of tree samples [18].

## 3. Results and Discussions

*3.1. Error Compensator*

All symbols and their descriptions refer to Table 1. The improved allometric equation $g_{(x)} = x^k e^{-ux}$ can be approximated at $x = \mu$ as

$$g_{(x)} \approx \sum_{q=0}^{2} [(1/q!)\, g^{(q)}{}_{(\mu)} (x - \mu)^q], \tag{1}$$

by using a 2nd-order Taylor series expansion and omitting the remainder term. Taking expectations on both sides, then we see that

$$\mu_g \approx E\{\sum_{q=0}^{2} [(1/q!)\, g^{(q)}{}_{(\mu)} (x - \mu)^q]\}, \tag{2}$$

After expanding the right side of Equation (2), and then substituting $g_{(x)} = x^k e^{-ux}$ into the expansion, it gives the following expression:

$$\mu_g \approx \mu^k e^{-u\mu} + 0.5\mu^{k-2} e^{-u\mu} (k^2 - 2ku\mu - k + u^2\mu^2)\sigma^2, \tag{3}$$

By substituting $\mu_g$ into $error = Ar\,(\mu_g - \mu^k e^{-u\mu})$, it leads to a simple form (also refer to Appendix A)

$$error \approx 0.5Ar\mu^{k-2} e^{-u\mu} (k^2 - k - 2ku\mu + u^2\mu^2)\sigma^2, \tag{4}$$

for any probability distribution. Then we have the simple form of the error per unit area as:

$$\Phi_{(\mu,\,\sigma^2)} = 0.5r\mu^{k-2} e^{-u\mu} (k^2 - k - 2ku\mu + u^2\mu^2)\sigma^2, \tag{5}$$

This is the compensator for correcting the scaling error. Thus, the ratio $|\Phi_{(\mu, \sigma^2)}|/(r\mu^k e^{-u\mu})$ of regional biomass density becomes:

$$\eta_{(\mu, \sigma^2)} = 0.5\, \mu^{-2}(k^2 - k - 2ku\mu + u^2\mu^2)\sigma^2, \tag{6}$$

**Table 1.** Symbols and their descriptions.

| Symbol | Description | Unit |
|---|---|---|
| A | Total forested area in a region, $A = \sum a_i$. | ha |
| $a_i$ | The area of $i$th stand. | ha |
| $C_j^s$ | The combination. $C_j^s = j!/[s!(j - s)!]$. $j$ represents the total number of elements, and $s$ is the number of elements being chosen at a time. | - |
| $D_m$ | The maximum variance, which is a function of expectation. | $m^6\ ha^{-2}$ |
| E | The mathematical expectation operator. | - |
| $g_{(x)}$ | The function of random variable x, $g_{(x)} = x^k e^{-ux}$. | - |
| $i$ | Stand number, $i = 1, 2, \ldots, n$. | - |
| $k$ | Parameter, $0 < k \leq 1$. | - |
| $n$ | The number of stands. | - |
| $q$ | Derivative order, $q$ is set up to 4 in this study. | - |
| $r$ | Parameter, $r > 0$. | - |
| $u$ | Parameter, $u > 0$. | $m^{-3}\ ha$ |
| $w_i$ | Weight or the probability of occurrence of ith stand, $w_i = a_i/A$. | - |
| x | Stocking volume of a fine-scale area, e.g., a stand. | $m^3\ ha^{-1}$ |
| $x_i$ | Stand stocking volume of $i$th stand. | $m^3\ ha^{-1}$ |
| $x_{max}$ | The maximum possible value of x in the forest inventory. | $m^3\ ha^{-1}$ |
| $x_{min}$ | The minimum possible value of x in the forest inventory. | $m^3\ ha^{-1}$ |
| $Y_1$ | Regional total biomass ideally supposed to accumulate from all stands. | Mg |
| $Y_2$ | Regional total biomass calculated from A and $\mu$, $Y_2 = Ar\mu^k e^{-ux}$. | Mg |
| $y_i$ | Stand biomass density of $i$th stand or sample plot, $y_i = r\, x_i^k\, e^{-ux}$. | $Mg\ ha^{-1}$ |
| $z$ | Intermediate parameter (0~1), the percentage of the samples at the point of $x_{max}$ to total samples. | - |
| $\Phi_{(\mu, \sigma^2)}$ | The compensator of scaling-up error. | $Mg\ ha^{-1}$ |
| $\eta_{(\mu, \sigma^2)}$ | The compensation rate; $\eta_{(\mu, \sigma^2)} = \Phi_{(\mu, \sigma^2)}/(Y_2/A) = \Phi_{(\mu, \sigma^2)}/(r\mu^k e^{-ux})$. | - |
| $\mu\,(x^s)$ | The expectation of $x^s$, $\mu_{(x^s)} = E_{(x^s)}$, $s = 1, 2$. | - |
| $\mu$ | The expectation of x. It denotes regional stocking volume. $\mu = \sum w_i x_i = E(x)$. | $m^3\ ha^{-1}$ |
| $\mu_g$ | The expectation of $g_{(x)}$, $\mu_g = E[g_{(x)}]$. | - |
| $\nu_s$ | sth-order central moment, $\nu_s = E[(x - \mu)^s] = \sum C_j^s\, \mu_{(x^j)}\, (-\mu)^{s-j}$, sigma from $j = 0$ to $s$, $s = 1, 2$. | - |
| $\sigma^2$ | Variance of random variable x. | $m^6\ ha^{-2}$ |

Rewrite it, we have:

$$\eta_{(\mu, \sigma^2)} = 0.5\mu^{-2}k(k - 1)\sigma^2 + u(0.5u - k\mu^{-1})\sigma^2, \tag{7}$$

where the domains are $x_{min} \leq \mu \leq x_{max}$, $0 \leq \sigma^2 \leq D_m$, and $0.8 \leq k \leq 1$ (in this study). Notes: (1) $D_m$ is derived by assuming that $x_i$ has the largest variance for a given expectation if and only if $x_i$ equals either $x_{min}$ or $x_{max}$; (2) $\mu_{(x^s)}$ and $\nu_s$ are used for calculating $\mu_g$.

For a distribution between $x_{min}$ and $x_{max}$, the definition of variance implies that a variance will be largest for a given expectation while all samples ($x_i$) are either $x_{min}$ or $x_{max}$. The maximum variance can be expressed as

$$D_m = -(\mu - x_{min})^2 + (\mu - x_{min})\,(x_{max} - x_{min}), \tag{8}$$

where $\mu$ represents regional stocking volume. This is for any probability distribution where $x_{min}$ and $x_{max}$ are known. Here $\mu$ and $D_m$ define a parabola going downwards in the $\mu - \sigma^2$ plane. $D_m$ becomes a boundary of the curved surface $\eta_{(\mu, \sigma^2)}$ (Figure 1). The derivation of $D_m$ is detailed in Appendix A.

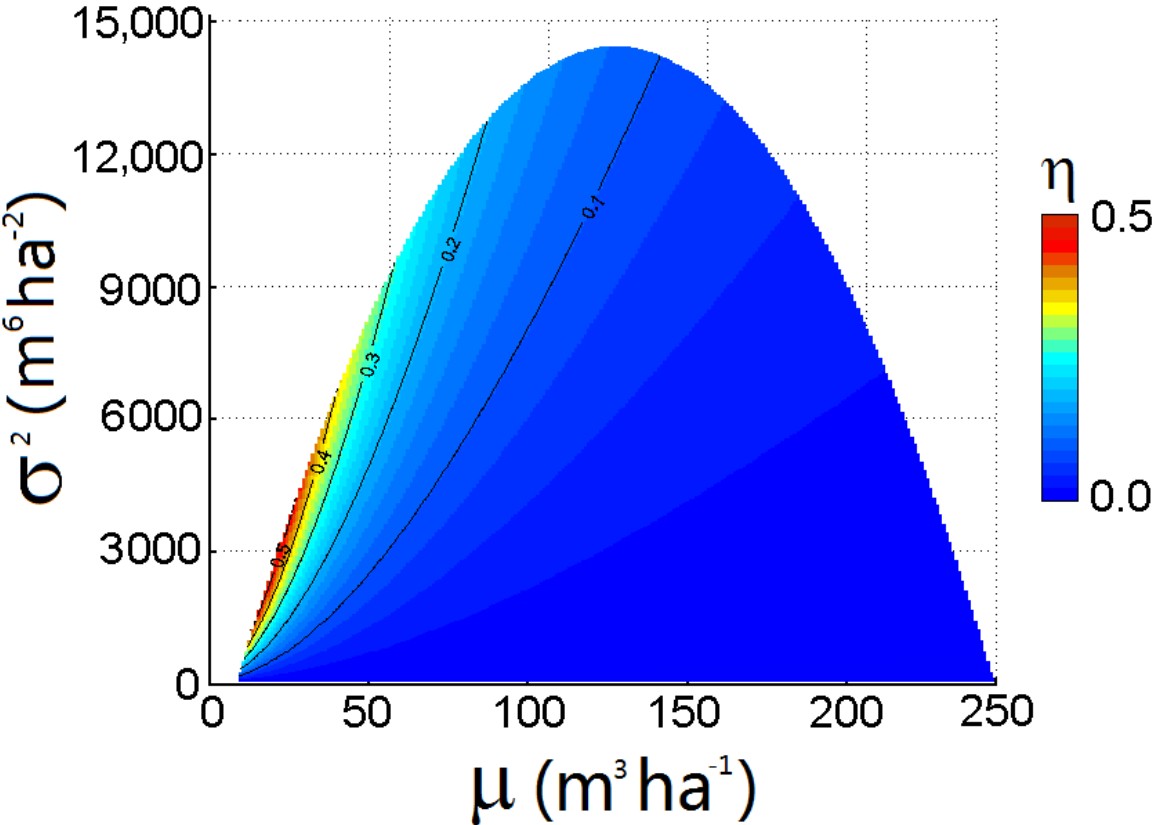

**Figure 1.** The percentage of scaling error compensated for *Pinus yunnanensis* Franch. η is the error compensation percentage given by Equation (7), in which this example uses parameters $r = 2.7468$, $k = 0.7623$ and, $u = 0.000536$. $\mu$ is the weighted mean of $x_i$ (stand stocking volume), and is also the mean regional volume. It is assumed to range from $x_{min}$ (10 m$^3$ ha$^{-1}$) to $x_{max}$ (250 m$^3$ ha$^{-1}$). $\sigma^2$ is the variance of $x_i$.

### 3.2. Efficiency of Reducing the Error

The efficiency and ability of the compensator to reduce scaling error is tested by 5000 trials. The mean of total biomass amount (true value) and the mean of modelled biomass amounts in the pseudo region are 44.7 and 46.3 (t), respectively (Figure 2a). It shows an approximate 3.5% overestimate due to the scaling error. After compensating, around 94.8% of the scaling error is reduced on average. The average level illustrates that compensation rate tends to become stable with the increase of trials. When trials approach 4000 times, Figure 2b indicates a convergence. This implies that most of the scaling error should be removed using the compensator.

The parameters in Equation (7) can explain how the scaling error is removed for the VAR equation ($y = rx^k\,e^{-ux}$). Equation (7) contains two terms. The first is the same as the compensation rate for the typical allometric equation $y = rx^k$, and the second is an addition for applying to the VAR equation. Both k and u affect the extent of error compensation. Because the VAR equation has a larger curvature than the typical allometric equation [35], the absolute value of $\eta_{(\mu,\,\sigma^2)}$ generally increases after adding the second term $u(0.5u - k\mu^{-1})\sigma^2$ to the first term in Equation (7). It means that the scaling error is larger using VAR than typical allometric equation. In extraordinary instances for some case, the volume-to-biomass relationship was considered as linear [33]. This makes *u* and *k* become zero and 1.0 respectively, and no scaling error will be caused in large-scale biomass estimation.

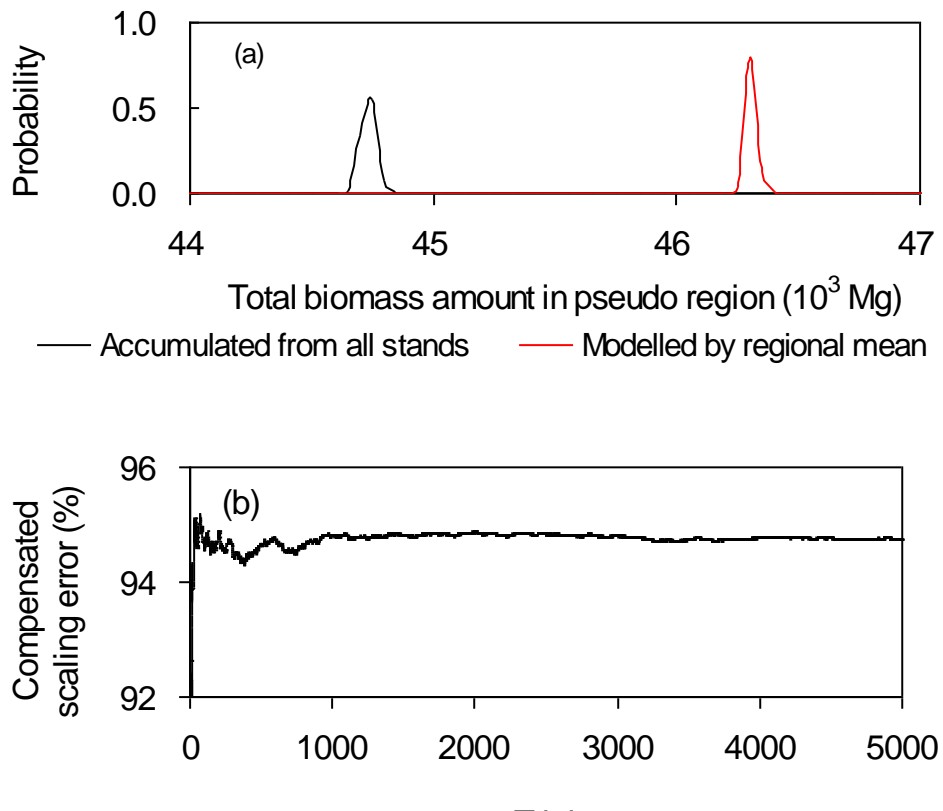

**Figure 2.** The efficiency of scaling error compensator based on statistic simulation. (**a**) Comparison between true value and modelled value of total amount of forest biomass in the pseudo region consisting of 100 stands (*n* = 100). (**b**) Percentage of compensated scaling error depending on trials. It becomes convergent after trials approach 4000 times. Note: the curve indicates the compensated scaling error in percentage, which is averaged from the trials.

*3.3. Comparison*

The results of the case study exhibit the percentage of scaling error that may occur from a biomass estimation of provincial scale forest ecosystem. The volume-to-biomass equation is parameterized using field measured data (Figure 3) for this estimation. After compensating scaling error, mean biomass density declines by 3.6% and becomes 69.1 (Mg ha$^{-1}$). This can result in a total forest biomass of 201.8 (10$^6$ Mg) for this species in the province (Table 2). The amount of reduced scaling error is 7.7 (10$^6$ Mg) using $\sigma^2 = 1721$ m$^6$ ha$^{-2}$. The fine dotted line (Figure 3) denotes the curve from the power-law equation. It cannot properly fit the measurements especially when the volume exceeds 200 m$^3$ ha$^{-1}$.

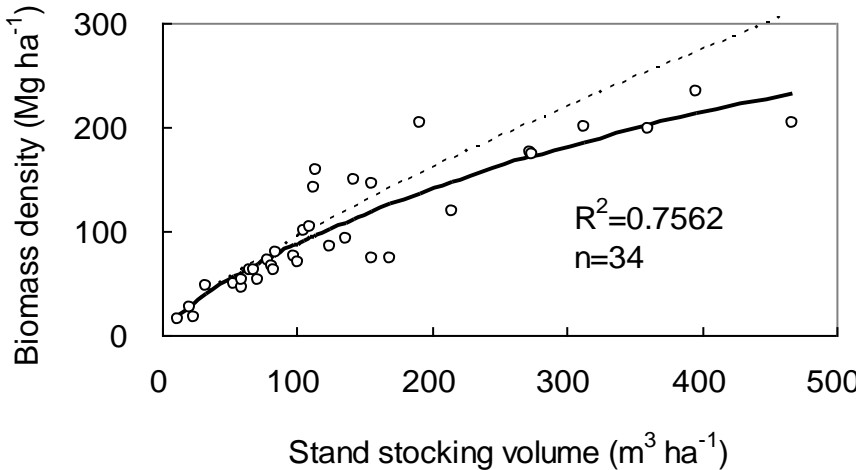

**Figure 3.** Stand stocking volume versus stand biomass density for parameterizing the volume-to-biomass equation $y = rx^k e^{-ux}$. The parameters estimated by nonlinear regression were $r = 2.7468$, $k = 0.7623$ and $u = 0.000536$ (m$^{-3}$ ha) for *Pinus yunnanensis*. The fine dotted line is a conventional power-law curve ($y = rx^k$, $r = 2.6526$, $k = 0.7742$, $R^2 = 0.7428$). Under the condition of the Jensen inequality $0 \le x \le (k + \sqrt{k})/u$, the range of x must be $0 \le x \le 3051$ (m$^{-3}$ ha). In practice, x is not greater than 500 (m$^{-3}$ ha). The condition is met.

**Table 2.** A case study of biomass estimation for *Pinus yunnanensis* at the provincial scale.

| | Unit | Young | Middle | Near-Mature | Mature | Over-Mature | Sum or Ave. |
|---|---|---|---|---|---|---|---|
| | | | | **Forest Age Groups** | | | |
| Total forested area A (data) | 10$^6$ ha | 1.051 | 0.955 | 0.480 | 0.341 | 0.096 | 2.92 |
| Total volume V (data) | 10$^6$ m$^3$ | 41.18 | 68.62 | 45.62 | 44.56 | 22.45 | 222.4 |
| Area proportion (p) | - | 0.360 | 0.327 | 0.164 | 0.117 | 0.033 | 1.0 |
| Mean stocking volume x and $\mu$ | m$^3$ ha$^{-1}$ | 39.2 | 71.9 | 95.1 | 130.8 | 233.8 | 76.1 |
| p(x − 76.1)$^2$ for calculating * | m$^6$ ha$^{-2}$ | 490.3 | 5.9 | 59.0 | 348.7 | 817.3 | 1721 |
| Mean biomass Y [†] | Mg ha$^{-1}$ | | | | | | 71.7 |
| Total biomass (71.7 × A) | 10$^6$ Mg | | | | | | 209.5 |
| Corrected Y | Mg ha$^{-1}$ | | | | | | 69.1 |
| Removable scaling error [‡] | % | | | | | | 3.6 |
| Total biomass (69.1 × A) | 10$^6$ Mg | | | | | | 201.8 |

* Variance is calculated based on stocking volume of each age group (data of A and V refer to the reference [38]).
[†] Biomass density is calculated using $y = rx^k e^{ux}$; parameters refer to the caption of Figure 3. [‡] Scaling error is calculated based on Equation (7).

The 3.6% overestimate is a remarkable percentage in this biomass estimation for the 2.92 (10$^6$ ha) forested area of *Pinus yunnanensis.* This error percentage is not necessarily for all species [22]. *Pinus yunnanensis* is a frigidly temperate coniferous forest species distributing from 23°1′ to 28°23′ N and 97°46′ to 105°54′ E in the province. It sustains drought and poor soils, widely adapts to extreme natural environmental conditions [41], and experiences fast growth on branch and leaves in middle-age [42,43]. This may result in its volume-to-biomass curve showing higher curvature than other species [23]. We found that the improved allometric VAR equation $y = rx^k e^{-ux}$ is more feasible than the conventional power-law equation for *Pinus yunnanensis*. This finding implies that the stand condition is decisive for the curvature of the compensator in Equations (5) and (7), including parameters $k$ and $u$. The values of parameters $k$ (0.7623) and $u$ (0.000536) are lower than the values for many species. This slowed growth under the poor site conditions is one of the possible reasons that the scaling error was reduced largely in the case study.

## 3.4. Uncertainty Analysis

As Equation 7 expressed, the estimate of variance $\sigma^2$ is a key variable that affects the bias of the compensator. To evaluate the variance and the detailed situation is difficult for large-scale forests [38]. If the variance is unknown, it is necessary to assume a distribution of the stocking volume, for instance,

uniform distribution. This would decrease the accuracy of compensation due to the bias of variance estimate. In the worst case, the variance bias could be $D_m - \sigma^2$, which can be derived as follows. We utilize $\sigma^2 = (x_{max} - x_{min})^2/12$ for calculating the variance of uniform distribution, and rewrite Equation (8) as $D_m = (\mu - x_{min})(x_{max} - \mu)$. The relative error of variance becomes

$$(D_m - \sigma^2)/D_m = (2/3)(\mu - x_{min})/(x_{max} - \mu), \tag{9}$$

As we have $\mu = (x_{min} + x_{max})/2$ for the uniform distribution, it leads to $(D_m - \sigma^2)/D_m = 2/3$. This implies that around 66% scaling error may be under-compensated by assuming a uniform distribution.

However, the worst condition is an extremity, in which the maximum variance $D_m$ is only for the distribution having all stands located at either $x_{min}$ or $x_{max}$. This stand structure does not exist in real forests. Practically, the distribution and forest structure are complex on tree height, stocking volume, and age group [39,44,45]. In our case study, we found that the distribution of stocking volume was close to an exponential curve on the area proportion (Table 2). Young stands have the largest area for *Pinus yunnanensis* [38]. Thus, the practical variance ($\sigma^2 = 1721$) is less than the variance of uniform distribution ($\sigma^2 = 3158$), which results in an estimate of $195.7 \times 10^6$ Mg. Comparing with the correct estimate ($201.9 \times 10^6$ Mg) of total biomass on true distribution, an over-compensation could occur. This implies that the uniform distribution may not be a proper assumption for some regions in which large afforestation and reforestation have been promoted for long term.

## 4. Conclusions

A large-scale analysis of forest ecosystems is usually based on fine-scale allometric relationships or equations. Depending on the nonlinear strength of the equations, a scaling (or aggregation) error may result, where regional applications using the stand level volume-to-biomass relationship need to conduct error compensation to detect and remove the possible scaling error.

Our simulation indicates that, on average, approximately 94.8% of the scaling error can be reduced. We have shown that significant overestimates of biomass (~3.6%) can be removed from a provincial estimate for *Pinus yunnanensis*. However, various allometric equations will need different error compensators. We suggest assessing the effects of scaling errors on large-scale estimations of spatial and temporal distribution of forest biomass and carbon.

**Supplementary Materials:** The following are available online at http://www.mdpi.com/1999-4907/10/7/602/s1, Supplementary Material S1: Supporting Information.

**Author Contributions:** C.Z. contributed to statistical analysis, simulation, and manuscript preparation. X.Z. conceived and designed experiment.

**Funding:** This work was supported by The National Key Research and Development Program of China (2017YFA0604401, 2016YFC0501101), Open Fund of State Key Laboratory of Remote Sensing Science (OFSLRSS201704) and Meteorology Scientific Research Fund in the Public Welfare of China (GYHY201506010).

**Acknowledgments:** The authors thank Joseph Carnegie and Jorge Potel for their helpful and valuable comments and suggestions. The authors thank Mingxia Yang for processing data and figures. The authors are also grateful to the three anonymous reviewers and the corresponding editors for their insightful comments and helpful suggestions.

**Conflicts of Interest:** The authors declare no conflict of interest.

## Appendix A.

*Appendix A.1. Problem Background of Scaling Error*

*(a) Example*

Supposition 1: A region includes two stands. The total area and volume are known for every stand (2 ha, 200 m$^3$; 3 ha, 100 m$^3$). If an allometric function is y = 2.7x$^{0.7}$ e$^{-0.0005x}$ where x denotes stocking volume, we calculate the regional biomass as follows:

$$Y_1 = 2[2.7(200/2)^{0.7}e^{-0.0005(200/2)}] + 3[2.7(100/3)^{0.7}e^{-0.0005(100/3)}] = 221.8 \ (\text{Mg ha}^{-1}), \qquad (A1)$$

Supposition 2: Only total area and volume are known for the region (5 ha, 300 m$^3$). The regional biomass may be calculated as

$$Y_2 = 5[2.7(300/5)^{0.7}e^{-0.0005(300/5)}] = 230.1 \ (\text{Mg ha}^{-1}), \qquad (A2)$$

This is an erroneous result, which shows an over-estimate by disregarding scaling error.

*(b) Analysis*

The improved allometric equation was presented as y = rx$^k$ e$^{-ux}$ [35] where *r*, *k*, and *u* are non-negative parameters. According to the actual situation, let x > 0 and $k < 1.0$. To apply the classical Jensen inequality for deriving scaling error, the function y must be a concave function. If letting g$_{(x)}$ = x$^k$e$^{-ux}$ and y = rg$_{(x)}$, the second derivative of g$_{(x)}$ should be equal to or less than zero, i.e., $u^2x^2 - 2kux - k + k^2 \leq 0$. Then we can obtain the condition to apply the Jensen inequality: $0 \leq x \leq (k + \sqrt{k})/u$.

The total biomass can be calculated through two different ways (Y$_1$ and Y$_2$):

$$Y_1 = \sum a_iy_i = \sum a_i(rx_i{}^k \ e^{-uxi}), \qquad (A3)$$

$$Y_2 = Ar\mu^k \ e^{-ux}, \qquad (A4)$$

where a$_i$ represents area of *i*th stand, and x$_i$ denotes total stand stocking volume (both variables are unknown); A is region area including numerous stands, and $\mu$ represents mean total stocking volume of an entire region. A and $\mu$ are known variables that can be obtained easily from the forested area and total volume provided by the inventories.

For computing, the first way Y$_1$ is precise and correct without scaling error since Y$_1$ is added up from all stands. However, this expression is not applicable since a$_i$ and x$_i$ are unknown for each single stand in a large region. Therefore, the second way Y$_2$ becomes the only choice to convert large-scale biomass directly from the mean stocking volume (m$^3$ ha$^{-1}$). We can rewrite Y$_1$ and Y$_2$ as follows:

$$Y_1 = \sum a_iy_i = \sum [A(\tfrac{a_i}{A})y_i] = \sum [Aw_i \ (rx_i{}^ke^{-uxi})] = Ar\sum [w_i(x_i{}^ke^{-uxi})], \qquad (A5)$$

$$Y_2 = Ar\mu^k \ e^{-ux} = Ar(\sum w_ix_i)^k \ e^{-u\sum wixi}, \qquad (A6)$$

where the random variable x is discrete with w$_i$. According to convex inequalities, e.g., weighted Jensen inequality, we have $\sum f(w_ix_i) \leq f(\sum w_ix_i)$ and $Y_1 \leq Y_2$, i.e., an error (*error* = Y$_1$ − Y$_2$) exist. this error is caused by scaling forest biomass from stand-scale up to large-scale. To solve the error, let $\mu_g = \sum[w_i(x_i{}^ke^{-uxi})]$, it leads to *error* = Ar ($\mu_g - \mu^k \ e^{-u\mu}$). Hence the issue becomes to find the unknown $\mu_g$ using known $\mu$ and $\sigma^2$. The step-by-step process is detailed in the results section.

*Appendix A.2. Derivation of $D_m$*

　　　Assuming *n* samples move from $x_{min}$ to $x_{max}$ one by one, the samples that are present at the point of $x_{min}$ and $x_{max}$ at any time are $n - nz$ and $nz$ respectively. Further, *z* is the percentage of the samples that have reached the point of $x_{max}$. Thus, we have

$$\mu = [(n - nz)x_{min} + nz\,x_{max}]/n, \tag{A7}$$

$$\mu_{(x^2)} = [(n - nz)(x_{min})^2 + nz(x_{max})^2]/n, \tag{A8}$$

Simplifying them leads to:

$$\mu = (1 - z)x_{min} + z\,x_{max}, \tag{A9}$$

$$\mu_{(x^2)} = (1 - z)(x_{min})^2 + z(x_{max})^2, \tag{A10}$$

Then we convert Equations (A9)–(A11):

$$z = (\mu - x_{min})/(x_{max} - x_{min}), \tag{A11}$$

Substitute Equations (A10) and (A11) into the variance formula $\sigma^2 = \mu_{(x^2)} - \mu^2$, we have:

$$\sigma^2 = -(\mu - x_{min})^2 + (\mu - x_{min})(x_{max} - x_{min}), \tag{A12}$$

Thus, the maximum variance is expressed as:

$$D_m = -(\mu - x_{min})^2 + (\mu - x_{min})(x_{max} - x_{min}), \tag{A13}$$

where $D_m$ is the boundary of variance. represents the regional stocking volume. This is for any distribution where $x_{min}$ and $x_{max}$ are known.

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
