# Peer review of "Removing the Scaling Error Caused by Allometric Modelling in Forest Biomass Estimation at Large Scales"

_forests, doi:10.3390/f10070602_

Reviewer 1 Report

General comment

After the revision, the manuscript has been substantially improved. So I recommend the publication of the manuscript after some very minor comments that I suggest below

Line 229

2.92 (106 ha-1) forested area -> it is 2.92 106 ha, please erase the -1 superscript

Line 263

… exist in realistic forests -> … exist in real forests

Line 265

… of stocking volume close to … -> … of stocking volume was close to …

Line 410

The area and volume… -> The total area and volume…

very

Line 428

… and xi denotes stand stocking… -> … and xi denotes total stand stocking…

Line 429

… and μ represents  mean stocking volume… -> … … and μ represents  mean total stocking volume…

Line 435

Mean stocking volume. The term: mean stocking volume means m3 ha-1 in terms of units. Please clarify here if it is the mean per ha value of stocking volume or the total stocking volume

Line 442

Hence the issue becomes to find unknown… -> Hence the issue becomes to find the unknown…

Line 443

… is detailed in the results section

Author Response

To the first reviewer:

>>> We appreciate Reviewer 1 for the positive feedback and helpful suggestions. Corresponding to the reviewer's comments, we have made revisions to improve our manuscript. The explanations are detailed as follows.

Line 229

2.92 (106 ha-1) forested area -> it is 2.92 106 ha, please erase the -1 superscript

>>> Thank you for pointing out this typo. We have corrected it.

Line 263

exist in realistic forests -> … exist in real forests

>>> Yes, we have replaced “realistic” with “real”.

Line 265

of stocking volume close to … -> … of stocking volume was close to …

>>> Yes, we have added “was” in the sentence.

Line 410

The area and volume… -> The total area and volume…

>>> Yes, we have added “total” in the sentence.

Line 428

and xi denotes stand stocking… -> … and xi denotes total stand stocking…

>>> Yes, we have added “total” in the sentence.

Line 429

and μ represents mean stocking volume… -> … … and μ represents  mean total stocking volume…

>>> Yes, we have added “total” in the sentence.

Line 435

Mean stocking volume. The term: mean stocking volume means m3 ha-1 in terms of units. Please clarify here if it is the mean per ha value of stocking volume or the total stocking volume

>>> Thank you very much for this good suggestion! We have added “(m3 ha-1)” to clarify that it is the mean hectare-based value of stocking volume.

Line 442

Hence the issue becomes to find unknown… -> Hence the issue becomes to find the unknown…

>>>Yes, we have added “the” in the sentence.

Line 443

is detailed in the results section.

>>> Yes, we have improved it as suggested

Reviewer 2 Report

Please see attached pdf file

Author Response

To the second reviewer:

The revised manuscript has correctly addressed my main comments. The added section on “Uncertainty analysis”, in particular, is much useful. The revised Table 2 now makes me understand that s2 for Pinus yunnanensis is computed as the between-group variance (where groups are based on forest age) rather than as the total variance (sum of the between- and within-group variance). What is the reason for disregarding the within-group variance? This should be clarified in the “Data for case study” Section. I have a few more minor comments that are listed below.

>>> We appreciate Reviewer 2 for the positive feedback and helpful suggestions. Corresponding to the reviewer's comments, we have added two sentences in the section of “Data for case study” to explain the age group and the group variance. The within-group variance could not be computed because the detailed data are not released to public in China.

L90-94, “Specifically (…) error compensator”: these lines are no longer useful given the text added at lines 71-74 and 83-84, so I would suggest to drop them. Moreover, saying that “tree volume” is the predictor is quite confusing since the predictor in this study is actually the plot-level volume.

>>> Yes, we totally agree with the reviewer's comments. These lines have been deleted.

L127, “34 measurements”: do you mean “34 plot measurements”? Please clarify.

>>> Yes, it means 34 plot measurements. We have added “34” into the sentence.

L127, “24 literatures”: what do you mean by “literature”? It was not clear to me.

>>> It means “24 studies”. We have replaced “literature” with “studies”.

L131-132: I would suggest to drop this sentence. What matters is the plot-level estimates of volume and biomass.

>>> We agree with the reviewer's comments. This sentence has been deleted.

L136: the second derivative of g exists everywhere, so you do not have to assume it (“Assuming that…” ->”Because…”)

>>> We appreciate this good suggestion. We have deleted the sentence for the brevity of the description.

L149, Eqn.2: e–uX -> e–uμ

>>> Thank you for pointing out this typo. We have corrected it.

P8, Table 1: the unit for all volume variances (i.e. Dm and s2) are m6 ha–2, not m6 ha–1.

>>> Thank you for pointing out this typo. We have corrected it.

L167: you may say that μ and Dm define a parabola in the μ-s2 plane or that they define a parabolic cylinder in the μ-s2-h space, but you cannot say that they define a parabolic cylinder in the μ-s2 plane.

>>> Yes, we totally agree with the reviewer's comments. We have added a sentence of “Here m and Dm define a parabola going downwards in the m-s2 plane”.

P9, Figure 1: I appreciated the effort to clarify the figure but must say that it is still quite illegible to me. Why using a 3D perspective that it is difficult to visualize when colours would allow you to plot the surface in a 2D plane? I would recommend to plot the surface as a 2D image.

>>> This is a very good suggestion! After trying plot a 2D figure, we feel that it is indeed easier to understand than before. We have replaced Figure 1 with a 2D image

Moreover, please consider that h as defined by Eqn.3 is negative while the scale given in Figure 1 shows positive values. Please be consistent.

>>> We are grateful to the reviewer for pointing out this mistake. We have used absolute value sign (|F(m, s2)|; see line 145 in the revised manuscript) to keep consistent between Eqn.3 and Figure 1

L117, L183 and Figure 2(b): the convergence is still unclear to me. Are you cumulating the data from one trial to the next one? I.e. the first trial consists of 100 simulated stands, the second trial consists of 200 stands (the 100 stands from the 1st trial plus 100 additional stands), etc. and the ith trial consists of 100i stands. Otherwise, I do not understand why the scaling error is converging. Please clarify. Please notice that Figure 2(b) is now OK on the screen but does not print correctly (at least with my printer).

>>> Yes, our basic thinking is similar to the cumulation, but it is a repeated averaging. The first trial consists of 100 stands (obtaining the first scaling error), the second trial consists of another 100 stands (obtaining the second scaling error), then we average these two scaling errors. In the third trial, we average all three scaling errors. Thus, the ith trial consists of the average values for 100i stands. To clarify this, we have added a note in the caption of Figure 2 as “Note: the curve indicates the compensated scaling error in percentage, which is averaged from the trials”. We hope that this will explain the computation. We are not very sure about the print problem for Figure 2(b). It may be because of the picture format. We have re-copied and pasted it again in the manuscript. Hope it works.

L217: please clarify “using s2 = 1721 m6 ha–2”.

>>> Yes, we have added “using s2 = 1721 m6 ha–2” in the sentence (line 213 in the revised manuscript).

P12, Table 2: you can remove the 8th row (“Variance s2”) since s2 is already given at the 5th row. The unit of s2 is m6 ha–2, not m6 ha–1 (and this unit should be shown in the 5th row: these are not dimensionless quantities).

>>> Yes, we have deleted the 8th row, and corrected the unit in Table 2

This manuscript is a resubmission of an earlier submission. The following is a list of the peer review reports and author responses from that submission.

Round  1

Reviewer 1 Report

Review Forests-498606

The manuscript reports a study on a procedure for removing “scaling error” caused from the allometric model in forest biomass estimation. The study is based on synthetic data and on a biomass model for Pinus yunnanensis in China. The paper is well written and statistically sound, nevertheless, it has some major conceptual flaws which sadly leads me to suggest its rejection. Please, see below the main concerns of this reviewer.

- From my point of view the authors misunderstand the meaning of “scaling error” in the forest inventory context. It has to do with the fact of aggregating estimations of trees, plots and stands to larger (population) scales. In this study, the allometric model considered estimates stand biomass density (y) as a function of stand volume density(x) which supposes that the hierarchical data structure (which is the main reason of the scaling errors…) is being ignored. Just as a side comment, the common procedure for transforming stand volume to stand biomass are the biomass expansion factors, being quite rare the use of the models proposed in the present study (see for instance the first line in Box 1).

- The way of defining the scaling error (see Box1) is strongly influenced by sampling error in practice, as the estimation of X (see Box 1 and Table 1…) is unbiased by definition, but random, and therefore the value of error = Y2-Y1 is a random (unknown) variable which cannot be corrected in any case. In this study, the sampling error in the determination of X is not considered in any case.

Reviewer 2 Report

Please see attached pdf file

Reviewer 3 Report

General comments:

It is true that the error reduction in the estimation of forest biomass using allometric models is a continuous task, especially when there are no direct biomass measurements and the stocking volume is used as raw data for biomass estimation.

The aim of this article was to publish a methodology using Taylor series of reducing the scaling error caused by the application of the standard allometric model for the estimation of forest biomass from tree volume data.

The article's main contribution is that it gives detail information on how the scaling error from allometric model can be reduced or removed in forest biomass estimation.

On the one hand, I found the manuscript to be overall well written and well described. I convinced that the authors performed careful and thorough work. The dataset used seems quite useful and sound for the purpose of the study. On the other hand, I found some parts on the methodology of the manuscript not clear. Therefore, I recommend that a minor revision is warranted. I explain below my concerns in more detail below and I ask the authors address my comments in their response.

Major comments:

I wander why there is a need to calculate compensator errors on regression equations that estimate tree or stand forest biomass from stocking volume once tree biomass it is already measured or estimated with some way?

If one wants, for example, to calibrate a VAR equation with compensation error must has somehow estimated or calculated tree biomass from the first place and if there are biomass data available then it is better to calibrate an allometric biomass model directly from biomass data rather than indirectly from stocking volume. What makes the scaling error removal approach useful is that it is more accurate than the biomass expansion factor (BEF) approach and can be applied in forest trees and stands at regional level where there are no biomass data available but stocking volume data from forest stewardship plans or forest inventories. I think that this must be addressed explicitly by the authors in the introduction section  

In the methods section, the main concern I have is with respect to the data for the case study, it is not clear to me how biomass raw data was derived and with which methodology for running the regression model presented in the section 3.3. If the citation N [33] refers to biomass estimation (I mean how biomass values were derived, by drying tree samples, or by some other model or equation) then this must be explicitly note it in the section 2.

Minor comments:

Line 22

In future earth environment studies -> In future earth environmental studies

Box 1.

After the reference Ruark et al. (1987) please write also the citation number in [ ]

, x can be stand volume density ->  , x can be stand for stocking volume

In forestry generally, and in growth and yield more specifically the proper scientific term is stocking volume which is expressed in m3 ha-1, so I’ d rather suggest to use the term stocking volume instead of volume density everywhere in the text

Line 86

, x can be stand volume density ->  , x can be stand for stocking volume

Line 99

… consisting of 100 stands with different sizes. -> … consisting of 100 stands with different stages of development

Line 149

Where X represents …  -> This symbol is different in table 1 please be consistent with the symbols used

SBD (what does SBD mean?) please write the whole words and SBD in parenthesis

Table 1

15th raw, small x Stocking volume… (why in t ha-1)? Stocking volume is measured in m3 ha-1

16th raw capital X same as previous comment (m3 ha-1 )

17th raw up to 19th raw again the symbols refer to stocking volume so the units must be (m3 ha-1 )

Lines 170 -171

Mind the units for xmin and xmax. Must be consistent with those in table 1

Box 3

Thus substitute Equation (A5) and (A6) into… -> Where are A4 A5 and A6, do you mean eqn 4 or  A4 A5 and A6 are the same with B4 B5 and B6?

where X represents the regional mean SBD. ->  In table 1 X sands for weighted mean of X or regional stocking volume? SBD stands for stand biomass density? It is not clear, please clarify

Line 227

2.92 (106 ha-1) forested area -> it is 2.92 (106 ha), please erase the -1 superscript

Figure 3 Caption

The parameters are determined as r=2.7468, -> I’ d rather suggest to change this with: The estimated by nonlinear regression parameters were: r=2.7468, ….

and other subtropical pines… -> What do you mean here?  Apart from P. yunnanensis the model was also calibrated for other subtropical pines? Which were they and bearing in mind that the parameters are the same did the Authors consider all those different pine species as one?

Line 228

Pinus yunnanensis  is a frigidly temperate coniferous forest… -> I’ d rather suggest:  Pinus yunnanensis  is a frigidly temperate coniferous forest species ….

Or Pinus yunnanensis forms frigidly temperate coniferous forests….

Line 230

…widely adapts to bad natural environment… -> I’ d rather suggest: …widely adapts to extreme  natural environmental conditions

Line 232

It is well known in growth and yield that site conditions have strong effect on stand increment especially in old aged stands the increment decreases substantially

Table 2

First raw, third column, the unit is ha not ha-1

Third raw, first column, write: Mean Stocking volume

Fourth raw, first column, write: Mean Biomass

In conclusion section, I’d rather suggest the Authors to write the conclusions in the form of bullets list, for example

● VAR allomteric equation is suitable for modelling growing under poor stand conditions

●Regional applicability…. Needs to contact error compensation to detect and remove scaling errors

● A significant overestimate of 6.27%  can be removed ….

……

● Approximately 94.8% of the scaling error reduces on average…

Lines 269-270

This sentence is not conclusions but rather methodology and results.